# Effects of *Centella asiatica* Extracts on Rumen In Vitro Fermentation Characteristics and Digestibility

**DOI:** 10.3390/ani14131956

**Published:** 2024-07-02

**Authors:** Yukyoung Yang, Kiyeon Park, Honggu Lee

**Affiliations:** Department of Animal Science, Konkuk University, Seoul 20249, Republic of Korea

**Keywords:** *Centella asiatica*, plant extracts, feed additive, methane

## Abstract

**Simple Summary:**

*Centella asiatica* has the potential to be utilized as a feed additive for ruminants because of its antiprotozoal, antibacterial, and antioxidant effects. However, studies evaluating *C. asiatica* as a feed additive are limited. Two experiments aimed to determine its optimal dosage and measure rumen degradability. The objective of the first experiment was to identify the effects of CAE on gas production and determine the optimal dosage of CAE as feed supplements. The CAE increased the total gas, methane, and total volatile fatty acid production. Here, 3.05% supplementation of CAE was chosen as the optimal dosage. The objective of the second experiment was to evaluate the effects of CAE on the rumen’s in vitro degradability. The in vitro dry matter, crude protein, and neutral detergent fiber degradability were not influenced by CAE supplementation. In conclusion, CAE has the potential to be utilized as a rumen fermentation-facilitating feed additive for ruminants.

**Abstract:**

Two in vitro experiments were conducted to evaluate the effects of *Centella asiatica* extract (CAE) supplementation on the rumen’s in vitro fermentation characteristics. In the first experiment, CAE with five concentrations (C: 0%; T1: 3.05%; T2: 6.1%; T3: 12.2%; and T4: 24.4% CAE in diet) was supplemented in the rumen fluid and incubated for 6, 24, and 48 h to determine the optimal dosage. The total gas and methane production increased in all incubation times, and the total volatile fatty acids increased at 6 and 48 h. Ammonia nitrogen, branched chain volatile fatty acids, acetate, and butyrate were increased by CAE supplementation. T1 was chosen as the optimal dosage based on the total volatile fatty acids, branched chain volatile fatty acids, and ammonia nitrogen production. The CAE with the identified optimal dosage (T1) was incubated to identify its effect on the rumen’s in vitro degradability in the second experiment. The CAE supplementation did not influence the in vitro dry matter, crude protein, or neutral detergent fiber degradability. In conclusion, CAE has no CH_4_ abatement or digestion promotion effects. However, CAE could be utilized as a feed additive to increase the rumen’s total volatile fatty acid production without an adverse effect on the in vitro dry matter, crude protein, or neutral detergent fiber degradability.

## 1. Introduction

Ruminants contribute a significant portion of the protein supply for humans [1]. Ruminants can utilize low-quality feeds and convert them into high-quality outputs through the vast population of microorganisms within the rumen and their symbiotic relationship with the host animal. Rumen microbes possess fiber-degrading enzymes, allowing them to decompose and utilize plant cell wall components. Along with this process, volatile fatty acids (VFAs), the major energy source for ruminants, are produced by rumen microbes. Microbes that escape the rumen are also an important source of protein for the host animal [2,3]. However, as a byproduct of rumen digestion, methane (CH_4_), a significant greenhouse gas, is emitted during this process. The livestock sector accounts for 14.5% of anthropogenic greenhouse gas emissions, and CH_4_ from the enteric fermentation of ruminant livestock accounts for 39.1% of greenhouse gas emissions from the livestock sector [4]. As CH_4_ is also energy-bearing organic matter, it accounts for 2–12% of the gross energy [5]. Therefore, diverse feed additives including plant extracts were studied to increase ruminant livestock productivity as well as alleviate global climate change [6,7,8]. Among these feed additives, plant secondary metabolites have been largely exploited due to their bioactive properties, including saponins [9], polyphenols [10,11], and flavonoids [12].

*Centella asiatica*, a medicinal herb that has been used in the orient, is known for its antiprotozoal [13], antibacterial [14], and antioxidant properties [15] along with many other therapeutic properties. Triterpenoids and saponins have been suggested to be the primary bioactive compounds of *C. asiatica*, including asiaticosides, madecassoside, and madasiatic acid [16]. Assuming that the bioactive compounds in *C. asiatica* could adjust the rumen’s microbial ecosystem to lower CH_4_ emissions and increase fermentation productivity, the use of *C. asiatica* powder in rumen fermentation was evaluated by Norrapoke et al. [17]. They reported the potential of *C. asiatica* as a methane-reductive additive based on the increased apparent fiber degradability and decreased acetate-to-propionate ratio (A:P), stoichiometrically calculated CH_4_ production, and protozoal population. These results seem incompatible because the potential methane-reductive mechanism of *C. asiatica* they mentioned was the effect of tannin and saponin on protozoa and protozoa-associated methanogen [18,19]. Likewise, the mixture of *C. asiatica* and mangosteen peel powder reduced the rumen protozoa count [17]. However, defaunation usually reduces fiber digestibility while reducing CH_4_ production [20]. On the other hand, feeding *C. asiatica* powder decreased CH_4_ production by only about 10 % compared with the control group, which is quite lower than other methane-reducing additives such as 3-nitrooxypropanol and *Asparagopsis taxiformis* [21,22,23]. Whether *C. asiatica* enhances NDF digestibility or reduces rumen CH_4_ emissions, we hypothesized that using *C. asiatica* extract (CAE) would maximize its effects and demonstrate its potential as a ruminant feed additive in terms of rumen fermentation and digestibility.

Two experiments were conducted to evaluate the effects of CAE on the rumen’s fermentation characteristics and digestibility. The objective of experiment 1 was to identify the effects of CAE on gas production and determine the optimal dosage of CAE as a feed supplement. The objective of experiment 2 was to evaluate the effects of CAE with the identified optimal dosage on the rumen’s in vitro degradability.

## 2. Materials and Methods

### 2.1. Ruminal Inoculum and Substrate

Three Holstein cows were used in this experiment. Rumen cannulae were installed according to the procedure approved by the Institution of Animal Care and Use Committee at Konkuk University (approval no. KU19087). Rumen fluid samples were collected from the cows’ ventral and dorsal sacs 2 h before morning feeding and filtrated into preheated thermos bottles using 250 μm pore-sized nylon filters. The rumen fluid obtained from each animal was mixed in the same proportion.

The donor animals were fed a commercial concentrate and roughage (tall fescue) mixed at a 4:6 (wt/wt) ratio, which was used as a substrate. Feed samples were air-dried for 24 h before being milled through a 1 mm screen for incubation. The contents of dry matter (DM), crude protein (CP), ash, and ether extract (EE) were analyzed in accordance with the AOAC’s official methods [24], and the contents of neutral detergent fiber (NDF) and acid detergent fiber (ADF) were analyzed using a neutral and acid detergent solution [25] containing sodium sulfite and heat-stable amylase. The non-fiber carbohydrate (NFC) portion was calculated as follows: 100 − (CP + EE + crude ash + NDF) (DM%). The chemical compositions of the diet are shown in Table 1.

### 2.2. C. asiatica Extraction

Dried *C. asiatica* powder (Byeongpul Farm Co., Ltd., Chungju-si, Republic of Korea) was extracted with a 60% ethanol solution for 16 h. The extracted solution was freeze-dried to remove the residual ethanol. Following extraction, the *C. asiatica* extracted powder was diluted with distilled water to a concentration of 200 mg/mL. The chemical compositions of the CAE are shown in Table 1.

### 2.3. Incubation Procedure

In experiment 1, McDougall’s medium [26] purged with CO_2_ was mixed with the filtrated rumen fluid at a 3:1 (vol:vol) ratio. The buffered rumen fluid was distributed to five flasks with different dosages of CAE (C: 0%; T1: 3.05%; T2: 6.1%; T3: 12.2%; and T4: 24.4% CAE in diet). The 30 mL of buffered rumen fluid was dispensed into 120 mL serum bottles which were then filled with 300 ± 1.5 mg of substrates. Following a flush with ultra-pure Ar (99.999%), the serum bottles were sealed [27,28]. Four replications for each treatment were incubated for 6, 24, and 48 h at 39 °C using a shaking incubator (JSSI-300C, JSR, Seongnam-si, Republic of Korea) set to 100 rpm. Then, 3 mL of a headspace gas sample was collected from all bottles at all time points using pre-vacutainer tubes with double septa (827B, Labco Ltd., Buckinghamshire, UK) with a gastight syringe (Gastight#1001, Hamilton Co., Reno, NV, USA).

In experiment 2, the medium of Menke [29] purged with CO_2_ was mixed with the filtrated rumen fluid at a 3:1 (vol:vol) ratio to reevaluate the possible intervention of the medium with the effects of CAE supplementation. The 50 mL of buffered rumen fluid was then dispensed into 200 mL bottles with two different dosages of CAE (0 and 3.05% CAE in rumen fluid). Each bottle had a fiber filter bag (filter bag 57; ANKOM Technology, Macedon, NY, USA) filled with 500 ± 25 mg of substrates. The bottles were sealed after flushing the headspace with ultra-pure Ar (99.999%). Four replications for each treatment were incubated for 24 h at 39 °C using a shaking incubator (JSSI-300C, JSR, Seongnam-si, Republic of Korea) set to 100 rpm.

### 2.4. Post-Fermentation Parameter Analyses

The post-fermentation parameters were analyzed as previously described by Park and Lee [27]. In brief, the total gas production (TGP) was calculated by measuring the headspace gas pressure [30] using a pressure transducer (Sun Bee Instrument Inc., Seoul, Republic of Korea). The methane concentration was measured by manually injecting 0.3 mL of headspace gas into a gas chromatograph (HP 6890 series GC system; Agilent Technologies Inc., Santa Clara, CA, USA) equipped with a thermal conductivity detector using a gastight syringe (Gastight#1001; Hamilton Co., Reno, NV, USA). The pH was measured using a pH meter (S20 SevenEasy pH; Mettler Toledo Co., Ltd., Greifensee, Switzerland). The VFA profile was analyzed using a gas chromatograph (HP6890 series GC system; Agilent Technologies Inc., Santa Clara, CA, USA) equipped with a flame ionization detector and a capillary column (DB-FFAP; Agilent Technologies Inc., Santa Clara, CA, USA). The ammonia nitrogen (NH_3_-N) concentration was analyzed with a catalyzed indophenol reaction [31] using spectrophotometry (Synergy2; Biotek Instruments, Inc., Winooski, VT, USA). The filter bags were thoroughly washed with cold water and air-dried at 60 °C for 48 h to measure the in vitro dry matter degradability (IVDMD). The NDF contents of the residual feeds in the bags were measured according to the procedure of Van Soest et al. [25] to measure the in vitro NDF degradability (IVNDFD). The CP contents of the residual feeds in the bags were measured in accordance with the AOAC’s official methods [24] using an elemental analyzer (EA 1110, CE instruments, Santa Clara, CA, USA) to measure the in vitro CP degradability (IVCPD).

### 2.5. Statistical Analysis

The data from the experiment 1 were analyzed using the GLM procedure in SAS OnDemand for Academics https://welcome.oda.sas.com/ (SAS Institute, Cary, NC, USA) with a fixed factor for the CAE supplementation level. The orthogonal contrast was conducted to investigate the linear and quadratic effects of CAE supplementation. All pairwise differences were separated by Tukey’s post hoc comparison. The data from experiment 2 were analyzed using an independent *t*-test in SAS OnDemand for Academics (SAS Institute, Cary, NC, USA). The significant differences were accepted if *p* < 0.05.

## 3. Results and Discussion

### 3.1. Experiment 1

The previous study conducted by Norrapoke et al. [17] reported that 2.5 DM% supplementation of *C. asiatica* powder did not increase the total VFAs but decreased the A:P ratio and stoichiometrically estimated CH_4_ production, possibly due to the anti-protozoal effects of tannin and saponin [17,18,19]. In addition, the CAE used in this study had a high percentage of NFCs, which primarily consist of non-structural carbohydrates, and a low percentage of NDF, which primarily consists of structural carbohydrates [32]. Given that the extraction method primarily extracted non-structural carbohydrates, it was hypothesized that the non-structural carbohydrates from the CAE would increase the propionate fermentation, which acts as the primary hydrogen sink in the rumen and indirectly inhibits methanogenesis, as methanogenesis is another major hydrogen sink in the rumen [33]. However, the propionate concentration decreased in all incubation times, and the A:P ratio and the CH_4_ emissions increased with the CAE in all incubation times (Table 2). Therefore, it was assumed that the ethanol extraction did not extract or denature the bioactive compounds that could reduce CH_4_ production, such as tannin and saponin. It was also assumed that the other compound from the CAE, included as NFCs but with structural carbohydrates such as pectin, facilitated acetate fermentation and increased methanogenesis in all incubation times [34]. Further studies are required to figure out which compounds from CAE induced the above results. 

Branched chain VFAs (BCVFAs) such as iso-butyrate and iso-valerate have been shown to be utilized by rumen microbes and be a key growth factor for certain rumen microbes, especially fiberlytic bacteria [3,35]. Therefore, the increase in ammonia caused by the addition of non-protein nitrogen resulted in a decrease in BCVFAs, which are used for microbial protein synthesis [36,37]. However, both NH_3_-N and BCVFAs were enhanced in all incubation times, which could be due to one of the following reasons: (1) protein degradation products were increased simply because the treatment groups had higher CP contents compared with the control group, or (2) unknown compounds in the CAE enhanced the rumen proteolysis rate. Whatever the reason was, the increased BCVFAs might have triggered fiber degradation because BCVFA supplementation can improve NDF digestibility [38,39]. This is supported by the increased acetate and TVFA production in this study. Experiment 2 was carried out to determine whether CAE improved the rumen degradability of DM, NDF, and CP as well as reevaluate the overall effects of CAE in different medium conditions. Considering the TVFA, BCVFA, and NH_3_-N results and cost-effectiveness, T1 was chosen as an optimal dosage for experiment 2. 

### 3.2. Experiment 2

Experiment 2 showed a decrease in the pH levels and an increase in TVFAs, which is consistent with the results for experiment 1 (Table 3). However, the IVDMD, IVNDFD, and IVCPD were not affected by the CAE. This indicates that CAE supplementation increased the extent of microbial fermentation, resulting in lower pH levels and more TVFAs by providing additional fermentable substrates from CAE, but it did not enhance the efficiency of the microbial fermentation, supported by maintained IVDMD, IVNDFD, and IVCPD. This is inconsistent with the work of Norrapoke et al. [17], who reported that *C. asiatica* powder supplementation increased apparent NDF and ADF digestibility. This could be due to the differences in supplementation type (powder vs. extracts), since it was assumed that the ethanol extraction did not extract or denature bioactive compounds from *C. asiatica*. 

On the other hand, NH_3_-N increasing with CAE supplementation could be due to the medium difference. In experiment 1, Mcdougall’s medium [26] was used, while Menke’s medium [29] was used in experiment 2. One of the main differences between the media was the additional nitrogen in Menke’s medium. Mcdougall’s medium is nitrogen-free, while Menke’s medium contains ammonium bicarbonate as an additional nitrogen source. This is supported by the overall elevated NH_3_-N level in experiment 2 compared with that of experiment 1. These findings suggest that the effect of CAE on rumen nitrogen degradation may be affected by the nitrogen levels in the rumen. The CAE increased the NH_3_-N concentration and BCVFA proportion in experiment 1 when the rumen microbes had a low level of available nitrogen. However, the higher nitrogen in the medium of experiment 2 influenced the effects of CAE on the rumen NH_3_-N concentration and iso-butyrate proportion, resulting in the maintained IVDMD, IVNDFD, and IVCPD. We suggest that there is an interaction between CAE and available nitrogen for rumen microbes. This could be determined in future studies using a factorial design study with CAE and non-ammonia nitrogen supplementation as treatments. In spite of the insignificant effects of CAE on rumen degradability, it increased TVFA, acetate, and butyrate production, which is consistent with the results obtained in experiment 1. On the other hand, the unchanged IVCPD implies that CAE did not enhance the proteolysis rate. Therefore, the increased proteolysis products in experiment 1 could be due to the higher nitrogen contents in CAE. 

## 4. Conclusions

The present study investigated the potential of CAE as a ruminant feed additive in terms of rumen VFA production, CH_4_ emissions, and digestibility. Although there were no differences in the rumen CH_4_ emissions and in vitro degradability, it was concluded that CAE could enhance rumen fermentation by increasing TVFA production. Thus, CAE can be utilized as a feed additive to facilitate rumen fermentation activity without adverse effects on rumen fermentation.

## Figures and Tables

**Table 1 animals-14-01956-t001:** Chemical composition of diet and *Centella asiatica* extract (percentage, dry matter basis).

Ingredient	CP	EE	Crude Ash	NDF	ADF	NFC
Tall fescue	5.53 ± 0.52	0.97 ± 0.52	6.43 ± 1.45	75.05 ± 3.56	46.32 ± 0.57	12.03 ± 2.11
Concentrate	14.04 ± 1.35	3.62 ± 0.35	7.98 ± 0.42	28.72 ± 6.94	13.81 ± 2.34	45.66 ± 8.22
CAE	3.04 ± 0.01	0.65 ± 0.00	16.54 ± 0.03	0.35 ± 0.00	0.23 ± 0.00	79.42 ± 0.05

CAE = *C. asiatica* extract; CP = crude protein; EE = ether extract; NDF = neutral detergent fiber; ADF = acid detergent fiber; NFC = non-fiber carbohydrate. Data are shown as mean ± standard deviation.

**Table 2 animals-14-01956-t002:** Effects of 0, 3.05, 6.1, 12.2, and 24.4% *Centella asiatica* extract supplementation on rumen fermentation characteristics after 6 h, 24 h, and 48 h of incubation (Experiment 1).

Items ^1^	*Centella asiatica* Extract Dosage ^2^					RMSE	*p* Value	Contrast ^3^	
	C	T1	T2	T3	T4			L	Q
6 h									
pH	6.94 ^a^	6.80 ^b^	6.77 ^b^	6.77 ^b^	6.76 ^b^	0.043	<0.001	<0.001	<0.001
Gas production (mL)	24.24 ^c^	30.72 ^ab^	32.59 ^ab^	30.37 ^b^	33.66 ^a^	1.476	<0.001	0.001	<0.001
CH_4_ (mL)	2.20 ^b^	4.45 ^a^	4.41 ^a^	4.29 ^a^	4.55 ^a^	0.191	<0.001	<0.001	<0.001
CH_4_ (%)	9.03 ^b^	14.4 ^a^	13.55 ^a^	14.16 ^a^	13.53 ^a^	0.569	<0.001	<0.001	<0.001
NH3-N (mg/dL)	0.32 ^b^	1.10 ^ab^	1.55 ^a^	0.97 ^ab^	1.28 ^ab^	0.471	0.025	0.257	0.057
Total VFA (mM)	64.12 ^c^	67.95 ^b^	74.48 ^a^	74.12 ^a^	73.93 ^a^	1.522	<0.001	0.719	<0.001
(mM/100 mM)									
Acetate	56.33 ^b^	59.11 ^a^	59.99 ^a^	59.72 ^a^	59.7 ^a^	0.692	<0.001	0.009	<0.001
Propionate	32.51 ^a^	27.56 ^b^	27.51 ^bc^	27.29 ^bc^	26.87 ^c^	0.305	<0.001	<0.001	<0.001
Iso-butyrate	0.58 ^b^	0.72 ^a^	0.71 ^a^	0.69 ^a^	0.71 ^a^	0.021	<0.001	<0.001	<0.001
Butyrate	7.89 ^b^	9.77 ^a^	9.71 ^a^	9.56 ^a^	9.89 ^a^	0.476	<0.001	0.001	<0.001
Iso-valerate	0.62 ^b^	0.96 ^a^	0.93 ^a^	0.92 ^a^	0.97 ^a^	0.033	<0.001	<0.001	<0.001
Valerate	2.06	1.87	1.80	1.82	1.85	0.134	0.089	0.387	0.063
A:P ratio	2.27 ^b^	3.04 ^a^	3.09 ^a^	3.10 ^a^	3.15 ^a^	0.142	<0.001	<0.001	<0.001
BCVFA	1.20 ^b^	1.69 ^a^	1.64 ^a^	1.62 ^a^	1.69 ^a^	0.051	<0.001	<0.001	<0.001
24 h									
pH	6.71	6.67	6.63	6.60	6.57	0.096	0.293	0.808	0.037
Gas production (mL)	42.92 ^c^	49.09 ^b^	53.13 ^a^	52.17 ^ab^	54.82 ^a^	1.781	<0.001	0.035	<0.001
CH_4_ (mL)	5.65 ^b^	8.47 ^a^	8.73 ^a^	8.63 ^a^	9.13 ^a^	0.337	<0.001	<0.001	<0.001
CH_4_ (%)	13.15 ^b^	17.28 ^a^	16.44 ^a^	16.55 ^a^	16.66 ^a^	0.521	<0.001	<0.001	<0.001
NH3-N (mg/dL)	6.35 ^c^	28.47 ^b^	32.41 ^ab^	33.45 ^ab^	36.74 ^a^	3.475	<0.001	<0.001	<0.001
Total VFA (mM)	90.22 ^ab^	98.63 ^ab^	108.74 ^a^	108.72 ^a^	84.92 ^b^	10.225	0.015	0.569	0.800
(mM/100 mM)									
Acetate	56.68 ^b^	59.81 ^a^	60.34 ^a^	59.87 ^a^	60.25 ^a^	0.626	<0.001	<0.001	<0.001
Propionate	30.69 ^a^	25.14 ^b^	24.95 ^b^	25.2 ^b^	24.32 ^b^	0.522	<0.001	<0.001	<0.001
Iso-butyrate	0.86 ^b^	1.26 ^a^	1.24 ^a^	1.25 ^a^	1.30 ^a^	0.052	<0.001	<0.001	<0.001
Butyrate	8.84 ^c^	10.11 ^ab^	9.88 ^b^	10.08 ^ab^	10.30 ^a^	0.180	<0.001	<0.001	<0.001
Iso-valerate	1.08 ^b^	1.90 ^a^	1.84 ^a^	1.85 ^a^	2.00 ^a^	0.098	<0.001	<0.001	<0.001
Valerate	1.85	1.80	1.75	1.75	1.83	0.087	0.377	0.994	0.626
A:P ratio	2.62 ^b^	3.37 ^a^	3.43 ^a^	3.37 ^a^	3.51 ^a^	0.079	<0.001	<0.001	<0.001
BCVFA	1.93 ^b^	3.15 ^a^	3.08 ^a^	3.10 ^a^	3.30 ^a^	0.147	<0.001	<0.001	<0.001
48 h									
pH	6.54 ^a^	6.49 ^ab^	6.48 ^ab^	6.50 ^ab^	6.46 ^b^	0.027	0.012	0.044	0.006
Gas production (mL)	50.47 ^c^	58.17 ^b^	59.59 ^ab^	58.43 ^b^	61.26 ^a^	1.087	<0.001	<0.001	<0.001
CH_4_ (mL)	7.38 ^b^	10.32 ^a^	10.34 ^a^	10.19 ^a^	10.68 ^a^	0.254	<0.001	<0.001	<0.001
CH_4_ (%)	14.61 ^b^	17.74 ^a^	17.35 ^a^	17.44 ^a^	17.43 ^a^	0.309	<0.001	<0.001	<0.001
NH3-N (mg/dL)	21.13 ^c^	46.31 ^bz^	48.75 ^ab^	47.98 ^ab^	52.48 ^a^	2.260	<0.001	<0.001	<0.001
Total VFA (mM)	103.73 ^b^	123.55 ^a^	122.20 ^a^	116.19 ^a^	115.36 ^ab^	5.663	0.001	0.004	0.296
(mM/100 mM)									
Acetate	57.06 ^b^	59.73 ^a^	59.96 ^a^	59.77 ^a^	59.64 ^a^	0.559	<0.001	<0.001	<0.001
Propionate	29.40 ^a^	24.76 ^b^	24.76 ^b^	25.06 ^b^	24.63 ^b^	0.448	<0.001	<0.001	<0.001
Iso-butyrate	1.13 ^bc^	1.45 ^a^	1.23 ^b^	1.08 ^cd^	0.98 ^d^	0.059	<0.001	<0.001	0.007
Butyrate	8.96 ^c^	9.98 ^b^	9.97 ^b^	10.00 ^b^	10.43 ^a^	0.117	<0.001	<0.001	<0.001
Iso-valerate	1.59 ^c^	2.26 ^b^	2.27 ^b^	2.26 ^b^	2.42 ^a^	0.050	<0.001	<0.001	<0.001
Valerate	1.87	1.82	1.82	1.82	1.91	0.074	0.394	0.852	0.391
A:P ratio	2.76 ^b^	3.42 ^a^	3.44 ^a^	3.38 ^a^	3.43 ^a^	0.070	<0.001	<0.001	<0.001
BCVFA	2.72 ^c^	3.71 ^a^	3.50 ^b^	3.35 ^b^	3.40 ^b^	0.088	<0.001	<0.001	<0.001

^a, b, c, d^ different superscripts within a row indicate significant differences (*p* < 0.05); ^1^ VFA: volatile fatty acids; A:P ratio: acetate:propionate ratio; BCVFA: branched chain volatile fatty acids. ^2^ Dosage of added extract as follows: C = 0%; T1 = 3.05%, T2 = 6.1%; T3 = 12.2%; and T4 = 24.4% of diet. ^3^ Probability of linear (L) or quadratic (Q) effect from *Centella asiatica* extract level.

**Table 3 animals-14-01956-t003:** Effects of *Centella asiatica* extract in rumen in vitro on dry matter, neutral detergent fiber, and crude protein degradability after 24 h of incubation (experiment 2).

Items ^1^	*Centella asiatica* Dosage (%, DM)		SEM ^2^	*p* Value
	0	3.05		
pH	7.03	6.71	0.035	0.002
NH_3_-N (mg/dL)	53.80	40.87	3.168	0.007
Total VFA (mM)	33.62	51.18	0.568	<0.001
(mM/100 mM)				
Acetate	22.18	30.84	3.926	0.067
Propionate	11.19	15.18	2.352	0.087
Iso-butyrate	0.48	0.31	1.142	0.010
Butyrate	3.13	3.80	0.041	0.215
Iso-valerate	0.79	0.69	0.255	0.450
Valerate	1.36	2.16	0.059	0.005
BCVFA	1.27	1.00	0.187	0.122
A:P ratio	2.12	1.99	0.085	0.196
IVDMD	43.49	44.41	2.243	0.572
IVNDFD	10.76	12.03	1.727	0.515
IVCPD	49.57	52.81	2.387	0.224

^1^ VFAs = volatile fatty acids; A:P ratio = acetate and propionate ratio; BCVFAs = branched chain volatile fatty acids; IVDMD = in vitro dry matter degradability; IVNDFD = in vitro neutral detergent fiber degradability; IVCPD = in vitro crude protein degradability. ^2^ SEM = sum of error of means.

## Data Availability

Upon reasonable request, the datasets of this study can be made available by the corresponding author.

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
