# Peer review of "Effects of Centella asiatica Extracts on Rumen In Vitro Fermentation Characteristics and Digestibility"

_animals, 2024, doi:10.3390/ani14131956_

Round 1

Reviewer 1 Report

Comments and Suggestions for Authors

The authors presented the effect of Centella asiatica extract at different levels on rumen in vitro fermentation characteristics, which fell into the scope of the journal animals. However, the authors should deal with the following problems or comments before being considered for further review.

First, the experiment design is kind of incorrect, such as: experiment 1 was not in accordance with experiment 1 on sample weight, bottle volume and medium type

Second, why did not analyze CO2 and H2 at the same time ?

Last, the authos did not show the characteristics of rumen liquid used in experiments 1 and 2

More comments in detail can be available in the attached PDF file

Comments on the Quality of English Language

L176, this result was supported by

Reviewer 2 Report

Comments and Suggestions for Authors

Even though this manuscript is interesting, a several concern needs to be addressed before further consideration. Please see the specific comment in pdf. file.

Comments on the Quality of English Language

Minor editing of English language required

Author Response

Thank you for the review and comments.
Please see the attachment. 

Reviewer 3 Report

Comments and Suggestions for Authors

1- Line 23 CH4 should be CH4

2-Line 28 DM, CP, and NDF authors should mention the full name first then use the abbreviations please 

3-Line 29 methane should be CH4

4-Line 43 greenhouse gas should put the abbreviation once authors will use it (GHGs) the same for methane 

5-Lines 63-65 should be removed and can be put into the introduction please 

6-Introduction lack more details about the C. asiatica  and a plant ? 

6- line 94 what about the source of Dried C. asiatica powder ? please clarify 

7-Line 93 more details needs for 2.2. C. asiatica extraction ? 

8-In my opinion its very essential fro the authors to mention the main active components in the C. asiatica extraction, its will help specially in there discussion 

9-Line 100 CO2  should be CO

10-The authors did not mention about CH4 collection ? how many samples? and the time of collections ? 

11-The VFA analyzed according to whom? and how authors did the collection and the samples preparation ? Clarify please 

12-In the materials and methods no thing mention about the pH? 

13-Line 151- and line 174 bioactive compounds ? which ? Please Clarify ? 

Comments on the Quality of English Language

 Minor editing of English language required

Author Response

(The authors gave the same response as above.)

Round 2

Reviewer 1 Report

Comments and Suggestions for Authors

L162, 6 h, 24 h, 48 h

L192, why did not the author present the data of total gas production or methane in table 3 ?

Reviewer 2 Report

Comments and Suggestions for Authors

I am grateful that you have accepted my opinion into the manuscript. In addition, there are a few small comments that the writers need to fix.

Neither the Abstract nor the Introduction section presents the aims of Experiment 2 in a clear and concise manner. However, the authors only discuss rumen fermentation in the abstract section, and they do not discuss digestibility. Although the introductory section focuses solely on digestibility, it does not include any characteristics related to rumen fermentation. Throughout the whole document, please verify and make improvements.

Author Response

We appreciate your detailed review. 

The comments were addressed in lines #9-14, 23-28, and #62-70 in the revised manuscript.
Thank you.